# Correlation of *TIMP1-MMP2/MMP9* Gene Expression Axis Changes with Treatment Efficacy and Survival of NSCLC Patients

**DOI:** 10.3390/biomedicines11071777

**Published:** 2023-06-21

**Authors:** Jacek Pietrzak, Agnieszka Wosiak, Dagmara Szmajda-Krygier, Rafał Świechowski, Mariusz Łochowski, Milena Pązik, Ewa Balcerczak

**Affiliations:** 1Laboratory of Molecular Diagnostics, Department of Pharmaceutical Biochemistry and Molecular Diagnostics, BRaIN Laboratories, Medical University of Lodz, Czechoslowacka 4, 92-216 Lodz, Poland; 2Department of Thoracic Surgery, Copernicus Memorial Hospital, Medical University of Lodz, Pabianicka 62, 93-513 Lodz, Poland

**Keywords:** lung cancer, NSCLC, metalloproteinases, tissue inhibitors of metalloproteinases, lung cancer treatment, NSCLC survival

## Abstract

In the course of lung cancer, normal cells are transformed into cancerous ones, and changes occur in the microenvironment, including the extracellular matrix (ECM), which is not only a scaffold for cells, but also a reservoir of cytokines, chemokines and growth factors. Metalloproteinases (MMPs) are among the elements that enable ECM remodeling. The publication focuses on the problem of changes in the gene expression of *MMP2*, *MMP9* and tissue inhibitor of metalloproteinases (*TIMP1*) in the blood of NSCLC patients during therapy (one year after surgical resection of the tumor). The paper also analyzes differences in the expression of the studied genes in the tumor tissue, as well as data collected in publicly available databases. The results of blood tests showed no differences in the expression of the tested genes during therapy; however, changes were observed in cancerous tissue, which was characterized by higher expression of *MMP2* and *MMP9*, compared to non-cancerous tissue, and unchanged expression of *TIMP1*. Nevertheless, higher expression of each of the studied genes was associated with shorter patient survival. Interestingly, it was not only the increased expression of metalloproteinase genes, but also the increased expression of the metalloproteinase inhibitor (*TIMP1*) that was unfavorable for patients.

## 1. Introduction

In 2020, more than 2.2 million people worldwide were diagnosed with lung cancer, which was the second most common type of cancer in the population. However, this cancer was characterized by the highest death rate [1]. Most lung cancer cases are mainly related to environmental factors, with smoking considered as crucial. In highly developed countries, the number of cases is decreasing due to lifestyle changes, but in lower-income countries, an increasing trend has been observed. However, lung cancer is not diagnosed in people exposed to tobacco smoke only [2]. The most frequent type is non-small-cell lung cancer (NSCLC), occurring in about 85% of patients. There are two most common histological subtypes observed, i.e., adenocarcinoma and squamous cell carcinoma [3]. A significant percentage of patients with diagnosed lung cancer present mutations in *K-ras* and the *EGFR*, *ALK*, *BRAF*, *MEK*, *MET* and *TP53* oncogenes [4]. Their status is a marker for diagnosis and is also used in targeted therapy. Treatment of patients with NSCLC mainly includes tumor resection and, at an advanced stage, adjuvant chemotherapy or differentiated targeted therapy, also in cases that previously developed resistance [5]. Despite significant progress, the effectiveness of therapy is still insufficient. Therefore, there is a need for a deeper understanding of the molecular changes that lead to the development of lung cancer or prognosis assessment [6].

Cancer arises as an accumulation of mutations in genes responsible for the proliferation or apoptosis process, which leads to uncontrolled cell growth. Abnormal neoplastic cells are surrounded by normal stromal cells, e.g., fibroblasts; however, neoplasia significantly changes their functioning. A similar process is typical for wound healing or inflammation. This leads to an increase in angiogenesis, turnover of the extracellular matrix (ECM) and tumor cell motility [7]. Extracellular proteinases, including metalloproteinases, participate in microenvironmental changes induced by tumor development. The activity of metalloproteinases is not limited to the degradation of the ECM. These enzymes regulate the activity of other proteases, growth factors, cytokines, surface receptors or their ligands, e.g., VEGF, TGF-β, IGFs. Thus, they become an important factor balancing the functions of both normal and neoplastic cells. The activity of metalloproteinases modulates the formation of new blood vessels by releasing VEGF from the ECM, as well as cleaving IGF-binding proteins and making this factor available for the cells. In many tumors, the activity of metalloproteinases is disturbed [8]. The activity of MMPs depends on their gene expression levels as well as the amount of proenzymes secreted into the extracellular space and the activation of zymogens. Finally, it can be blocked by the activity of tissue inhibitors of metalloproteinases (TIMPs) [9].

The metalloproteinase family is divided into six subgroups, according to their structure and specificity for the digested protein substrate. One of the subgroups includes gelatinases MMP-2 and MMP-9 with the ability to degrade gelatin [10]; however, despite belonging to the same group, they differ in terms of structure and function [11]. The *MMP2* gene is constitutively expressed in many tissues, and it is located on chromosome 13. The *MMP9* gene is located on chromosome 20, and it is most often expressed by stimuli from various types of transcription factors, such as NFκB [12].

The activity of metalloproteinases depends largely on the balance between MMP concentration and tissue inhibitors of metalloproteinases. The disturbed balance in neoplastic disease results not only from altered gene or protein expression of metalloproteinases, but also from tissue inhibitors of metalloproteinases [8,12]. So far, four types of TIMPs (TIMP-1, TIMP-2, TIMP-3 and TIMP-4) have been identified. They are characterized by a lack of specificity for individual MMPs, except for MT1-MMP. TIMP1 is secreted by most normal cells and inhibits the activity of a number of metalloproteinases, including MMP-2 and MMP-9 [13].

The presented study aims to assess the change, during one year after diagnosis, in the expression of *TIMP1 mRNA* and two metalloproteinases involved in the neoplastic process, the activity of which is inhibited by TIMP-1 in non-small-cell lung cancer. The gene expression of *TIMP1*, *MMP2* and *MMP9* has been assessed in tumor tissue and blood. The results of the analysis were compared to various clinical data. This work attempts to comprehensively describe the changes in gene expression of *gelatinases* and *TIMP1* in NSCLC, as a result of cancer development and progression as well as a consequence of applied treatment.

## 2. Materials and Methods

### 2.1. Materials

The research was carried out on RNA isolated from peripheral blood lymphocytes and from neoplastic lung tissue obtained from **(N = 45)** patients with non-small-cell lung cancer. The blood was taken from the subjects at three time points—**before tumor removal surgery (N = 45)**, **one hundred days after surgery (N = 39)** and **one year after surgery (N = 27)**. The research material also included blood from healthy individuals **(N = 38)** obtained from the regional blood donation center. The results of NSCLC patients were compared to those of a **control group**. The research was conducted with the approval of the Bioethics Committee (No. RNN/87/16/EC and KE/952/22). The clinicopathological characteristics of the study group are presented in Table 1. The number of samples taken from the patients at various time points was different due to their death or absence at subsequent time points.

### 2.2. Methods

#### 2.2.1. RNA Isolation

Tumor tissue collected intraoperatively was placed in RNA*later* Stabilization Solutions Invitrogen (USA) and stored until isolation at −80 °C. In the collected blood samples, red blood cells were lysed with eBioscience 1X RBC Lysis Buffer (Invitrogen, Waltham, MA, USA) and centrifuged. RNA*later* Stabilization Solution (Invitrogen, USA) was added to the resulting pellet, and the mixture was stored at −80 °C. Total RNA from peripheral blood and cancer tissue of NSCLC patients were isolated using the Total RNA Mini Plus kit from A&A Biotechnology (Gdańsk, Poland) according to the manufacturer’s protocols. The concentration and purity of the isolated RNA were measured using the spectrophotometric method with an Implen NanoPhotometer (Munich, Germany). RNA samples whose A260/A280 absorbance ratio was between 1.7 and 2.0 were qualified for the study.

#### 2.2.2. Reverse Transcription Reaction

The obtained RNA samples were reverse transcribed to cDNA. The reaction was performed using High Capacity cDNA Reverse Transcription Kits from Applied Biosystems (Waltham, MA, USA). Each sample used for the RT reaction had a final RNA concentration of 0.005 µg/mL. The composition of the reaction mixture and the temperature conditions of the reaction were in accordance with the manufacturer’s recommendations. The cDNA obtained in this process was stored at −20 °C until further analysis.

#### 2.2.3. PCR Reaction

To assess the presence of cDNA in all tested samples, a PCR reaction was performed using PCR Super Master Mix (Biotools, Jupiter, FL, USA) for *GADPH* (housekeeping gene), according to the manufacturer’s protocol. The sequences of the primers used for amplification are listed in Table 2. Agarose gel electrophoresis was used to check the presence of 96 bp products in the tested samples after amplification and confirm the expression of the reference gene in all samples.

#### 2.2.4. qPCR

The real-time PCR method was used to perform the quantitative evaluation of the expression of genes encoding *TIMP1*, *MMP2* and *MMP9* in peripheral blood and cancer tissue samples from the patients with non-small-cell lung cancer and the control group. The quantitative amplification reaction was performed on a CFX Connect Real-Time PCR Detection System (BioRad, Hercules, CA, USA) with the nonspecific SYBR Green fluorescent dye. The reaction mixture for one sample was composed of 5 µL of iTaq Universal SYBR Green Supermix BioRad (USA), 0.35 µL of each primer (Table 2), and RNA-free water to a final volume of 9 µL. The reaction conditions were as follows: initial denaturation step at 95 °C for three minutes and 40 cycles with two steps, i.e., denaturation at 95 °C for 10 s and annealing with elongation at 58 °C for 30 s. The real-time PCR reaction was performed in three replicates. For each experiment, the test (*TIMP1*, *MMP2*, *MMP9*), reference genes (*GAPDH*) and negative control were amplified in parallel. The mean of the obtained Ct values for *GAPDH* and all tested genes were calculated. To assess the relative level of *TIMP1*, *MMP2* and *MMP9* expression, the ^ΔΔ^Ct method was used due to the similar efficiency of the qPCR of each gene [14].

### 2.3. External Database

The assessment of changes in the expression of *TIMP1*, *MMP2* and *MMP9* genes in lung cancer was supplemented with the analysis of available databases. Thus, it was possible to compare neoplastic tissue and normal tissue as well as to contrast the obtained results with a larger population. For this purpose, the data collected in the UALCAN database [15] were analyzed. Samples examined in the TCGA program (The Cancer Genome Atlas Program) were selected for comparison. These data were used to compare the expression of the studied genes between cancerous and normal lung tissue. Additionally, the data collected in the database was used to compare the expression of the studied genes against various clinical features (smoking status, gender, TNM).

The data collected in the Kaplan–Meier Plotter database [16] were used to assess the relationship between the expression of the studied genes in the tumor tissue and the length of survival. Tumor cases were analyzed, both without any division into histopathological subtypes and with NSCLC categorized as adenocarcinoma and squamous cell carcinoma.

### 2.4. Statistical Analysis

The statistical analysis was performed using Statistica 13.1 software (TIBCO, Palo Alto, CA, USA). The Shapiro–Wilk test was used to assess the compliance of the distribution of the tested feature with the normal distribution. In order to obtain the compliance of the relative expression coefficients of the studied genes with the normal distribution, they were transformed with the Box–Cox transformant. The Mann–Whitney *U*-test, Kruskal–Wallis and ANOVA tests and repeated measures ANOVA were used to assess the differences between the individual populations studied. The log-rank test was used to analyze survival rates in general populations. In all the subsequent calculations, the significance was set at the level of 0.05.

## 3. Results

### 3.1. Correlation between TIMP1, MMP2 and MMP9 Gene Expression in Blood and Lung Tumor Tissue

The result of the statistical analysis did not confirm any correlation between the expression of the studied genes (*MMP2*, *MMP9*, *TIMP1*) in blood and tumor tissue (Table 3). An interesting result was shown by the comparison between the expression of *MMP2*, *MMP9* and *TIMP1* solely in the blood or neoplastic tissue (Table 3). The development of the neoplastic process causes the modification of the ECM remodeling only within the tumor microenvironment. Furthermore, the conducted study confirmed the presence of a positive correlation between the expression of gelatinases and their inhibitor (Table 3).

### 3.2. Assessment of MMP2, MMP9 and TIMP1 Gene Expression during the Therapy

In the group of patients diagnosed with NSCLC, the change in *TIMP1*, *MMP2* and *MMP9* expression was analyzed at three time points of the therapeutic process: before surgery, 100 days after surgery and one year after surgery. The analysis did not show any differences in the relative expression level of the studied genes (***TIMP1***, *p* = 0.1458; ***MMP2***, *p* = 0.1549; ***MMP9***, *p* = 0.4667) in the blood 100 days and one year following resection of tumor tissue (Figure 1).

### 3.3. Comparison of MMP2, MMP9 and TIMP1 Gene Expression in the Blood Depending on Adjuvant Chemotherapy

After surgical resection of tumor tissue, some patients undergo adjuvant chemotherapy with cisplatin and vinorelbine or carboplatin and gemcitabine. Therefore, two groups of patients were distinguished: the first group included individuals whose treatment was limited to surgical removal of neoplastic tissue (N = 27), and the second those who also received chemotherapy (N = 18). Both groups were compared for changes in the expression of all the tested genes at three time points. However, statistical analysis did not show any difference between the two groups (*TIMP1*, *p* = 0.9051; *MMP2*, *p* = 0.5377; *MMP9*, *p* = 0.7479). The results of the analysis indicate that chemotherapeutic treatment has no effect on the expression of any of the studied genes (Figure 2).

### 3.4. Comparison of MMP2, MMP9 and TIMP1 Gene Expression in the Blood between the NSCLC Patients and the Control Group

Statistical analysis showed no differences in the expression of the genes between the individuals with lung cancer at three time points and the control group (***TIMP1*** (before surgery *p* = 0.1083; 100 days after surgery *p* = 0.8224; one year after surgery *p* = 0.8457), ***MMP2*** (before surgery *p* = 0.938; 100 days after surgery *p* = 0.4236; one year after surgery *p* = 0.0527) and ***MMP9*** (before surgery *p* = 0.217; 100 days after surgery *p* = 0.5146; one year after surgery *p* = 0.278)).

### 3.5. The Changes in the Expression Level of TIMP1, MMP2 and MMP9 Genes in Blood at Three Time Points (before Surgery, 100 Days after Surgery and One Year after Surgery) against Various Etiological and Clinical Parameters

The results of the statistical analysis showed the impact of the tumor histological subtype on the change in *TIMP1* expression during the treatment period (*p* = 0.035). The main difference was observed 100 days after surgery. In adenocarcinoma, there was an increase in *TIMP1* expression 100 days after surgery compared to the baseline expression, while in squamous carcinoma, there was a decrease in *TIMP1* expression 100 days after surgery. Interestingly, one year after tumor removal, *TIMP1* expression was at a similar level in both histological subtypes of NSCLC. A similar tendency was observed for *MMP2* and *MMP9*, but it was not statistically significant. The study also assessed changes in the expression of all the genes in the blood of the NSCLC patients examined in the course of the treatment in relation to the size of the primary tumor according to the TNM classification. However, the results of the analysis did not show statistically significant changes in the expression of *TIMP1* (*p* = 0.6068), *MMP2* (*p* = 0.1589) or *MMP9* (*p* = 0.3996) in comparison with this parameter. In addition, no differences were found in the expression of all the studied genes at the analyzed time points depending on the presence of lymph node metastases (*TIMP1*, *p* = 0.9903; *MMP2*, *p* = 0.2114; *MMP9*, *p* = 0.4551) or histological grade (*TIMP1*, *p* = 0.8523; *MMP2*, *p* = 0.7353; *MMP9*, *p* = 0.9332). Moreover, the genes studied during the treatment were not affected by smoking status (*TIMP1*, *p* = 0.7116; *MMP2*, *p* = 0.8571; *MMP9*, *p* = 0.8463) or gender (*TIMP1*, *p* = 0.2795; *MMP2*, *p* = 0.8786; *MMP9*, *p* = 0.9958) (Figure 3).

### 3.6. Changes in the Expression of TIMP1, MMP2 and MMP9 in Neoplastic Tissue against Various Clinical Parameters

Due to the lack of correlation between the expression of all the studied genes in the blood and tissue of the NSCLC patients, a separate part of the analysis involved the assessment of changes in expression in tumor samples only. There were no differences in the expression of the studied genes in lung tissue according to tumor histological subtypes (*TIMP1* (*p* = 0.8535); *MMP2* (*p* = 0.4627) and *MMP9* (*p* = 0.0996)). The expression of the studied genes was at the same level in smokers and non-smokers (*TIMP1*, *p* = 0.6271; *MMP2*, *p* = 0.1866; *MMP9*, *p* = 0.2337). No correlation was found between the size of the tumor and the expression of the studied genes (*TIMP1*, *p* = 0.2465; *MMP2*, *p* = 0.8310; *MMP9*, *p* = 0.4369) in the neoplastic tissue, also taking into account their combined expression (*p* = 0.2698). Similarly, a lack of correlation was confirmed for the patients who had tumor cells in the surrounding lymph nodes as compared to those whose lymph nodes were free (*TIMP1*, *p* = 0.6652; *MMP2*, *p* = 0.5166; *MMP9*, *p* = 0.9766; collectively, *p* = 0.8416). No association between histological grade of malignancy and *TIMP1* (*p* = 0.8934), *MMP2* (*p* = 0.9332) or *MMP9* (0.9994) expression was confirmed.

### 3.7. Impact of TIMP1, MMP2 and MMP9 Gene Expression on Survival of NSCLC Patients

Survival analysis was performed based on the results of the expression of all the tested genes in neoplastic tissue and in blood prior to the surgical removal of the neoplastic tissue. The patients were divided into two groups based on the median value of the expression of individual genes (below the median, above the median). Additionally, a one-year survival analysis was performed based on the expression of the *MMP2*, *MMP9* and *TIMP1* genes in tissue and blood. The statistical analysis did not confirm the presence of the correlation between the expression of *TIMP1* (*p* = 0.4306), *MMP2* (*p* = 0.9208) or *MMP9* (*p* = 0.8674) in blood before surgical removal of tumor tissue and survival of patients with NSCLC. Similar results were obtained by analyzing the association between the expression of the studied genes in tumor samples and patient survival (*TIMP1*, *p* = 0.5972; *MMP2*, *p* = 0.9627; *MMP9*, *p* = 0.8563). The analysis of one-year survival in NSCLC patients showed no differences in the expression of the examined genes in the tumor tissue based on the one-year survival of the patients (*TIMP1* (*p* = 0.7463); *MMP2* (*p* = 0.8729); *MMP9* (*p* = 0.7338)) and in the blood taken before surgical resection (*TIMP1* (*p* = 0.5565); *MMP2* (*p* = 0.1253); *MMP9* (*p* = 0.304)). In addition, an analysis was carried out to assess the impact of the expression of the studied genes on overall survival, taking into account the differences between patients receiving adjuvant chemotherapy (at a more advanced stage of cancer) and those whose treatment procedure included only surgical resection of the tumor. However, the level of expression of *MMP2*, *MMP9* and *TIMP1* genes had no effect on overall survival in both study groups, in the blood of patients with adjuvant chemotherapy (*TIMP1* (*p* = 0.997); *MMP2* (*p* = 0.5153); *MMP9* (*p* = 0.7648)) and in the blood of patients without adjuvant chemotherapy (*TIMP1* (*p* = 0.1444); *MMP2* (*p* = 0.2631); *MMP9* (*p* = 0.7029)). Identical results were obtained for tumor tissue patients with adjuvant chemotherapy (*TIMP1* (*p* = 0.3108); *MMP2* (*p* = 0.6402); *MMP9* (*p* = 0.5738)) and without adjuvant chemotherapy (*TIMP1* (*p* = 0.2654); *MMP2* (*p* = 0.69); *MMP9* (*p* = 0.5649)).

### 3.8. Analysis of Data Collected in External Databases

The analysis of normal and neoplastic tissue, using data collected in the UALCAN database, showed a statistically significantly higher expression of the *MMP2* and *MMP9* (*p* < 0.0001) genes in NSCLC as compared to normal tissue; however, *TIMP1* gene expression remained unchanged. Similarly to the results of the experimental analysis, no differences were found in the expression of any of the tested genes between the histological subtypes of the tumor. The analysis did not confirm the effects of smoking status, gender, tumor size or surrounding lymph node involvement on the expression of the studied genes.

The patient survival analysis was also supplemented with an assessment of data collected in the Kaplan–Meier Plotter database. The analysis confirmed no correlation between the expression of *MMP2* and the survival of patients with lung cancer (*p* = 0.55). However, analysis of the specific histological subtypes of the tumor showed that in adenocarcinoma, higher expression correlates with shorter survival in NSCLC patients (*p* < 0.001), whereas for squamous cell carcinoma, no relationship was observed (*p* = 0.12). Similar results were obtained in the analysis of the length of survival of patients with NSCLC depending on *MMP9* expression. Without differentiation of the tumor subtypes, it has been shown that higher *MMP9* expression is associated with shorter survival in NSCLC patients. Assessment of a specific histological subtype, as in the case of *MMP2*, confirmed this relationship for adenocarcinoma only (*p* = 0.009), but not for squamous cell carcinoma (*p* = 0.34). The analysis of the relationship between *TIMP1* expression and survival time turned out to be very interesting. A statistically significant correlation was observed between lower *TIMP1* gene expression and longer survival of patients with lung cancer (*p* < 0.001). Analyzing the correlation between the expression of the *TIMP1* gene and the survival depending on a specific histological subtype of the tumor shows the correlation only in adenocarcinoma (*p* < 0.001). The results of survival analysis in NSCLC patients with adenocarcinoma are presented in Figure 4.

## 4. Discussion

The extracellular matrix (ECM) is present in all types of tissues as a three-dimensional non-cellular structure necessary for the proper functioning of cells, tissues, organs and ultimately whole organisms. The significance of this structure is reflected in the fact that various types of congenital abnormalities in ECM cause disorders that are fatal. The extracellular matrix not only provides the scaffolding in which cells are located, but also is the source of many different factors affecting the process of growth, migration, angiogenesis and survival [17]. Thus, it is an important element in various physiological and pathological processes, e.g., carcinogenesis. The extracellular matrix constantly undergoes remodeling, and it is one of the most dynamically changing structures of the body. Cells suspended in the ECM interact with it via surface receptors and integrate signals from the surrounding environment [18]. Moreover, cells are also responsible for the synthesis of various components of the ECM. Their great diversity affects not only the structure and biophysical properties, but also the signals that reach the cells, modulating their response. A number of proteolytic enzymes capable of degrading individual components, e.g., collagen, participate in the ECM transformation. Among them are metalloproteinases, proteolytic enzymes whose activity depends on the presence of the zinc ion in the active site [19]. These enzymes, through the degradation of the ECM components, are involved in various types of cellular processes. Changes in their concentration or activity are observed not only in inflammation, atherosclerosis and osteoarthritis, but also in the neoplastic process. The role of metalloproteinases in carcinogenesis is broad and crucial, especially in the early stages of development. Altered synthesis of MMPs or their inhibitors could change the activity of enzymes and affect the proliferation, apoptosis and angiogenesis of cancer cells as well as normal adjacent cells [20]. However, in the neoplastic process, there is no regular pattern of changes in the activity of individual metalloproteinases. Each cancer is characterized by a unique modification that reflects the type, stage and localization. The final pattern is the result of the interaction between the cancer cell and the environment, including the ECM, due to different changes in the activity of individual metalloproteinases [21]. An increase in the activity of some metalloproteinases may also be beneficial for cancer patients. MMP-9, despite its well-established role as a factor promoting the development of cancer, induces the expression of endostatin, which inhibits the formation of new blood vessels [22]. Finally, the diverse impact of metalloproteinases associated with the histological subtype of cancer, stage or location is confirmed by failures in research on inhibitors that were tested as universal anticancer drugs [23].

The presented study was primarily aimed to check the possibility of using the evaluation of *MMP2*, *MMP9* and *TIMP1* gene expression in the blood of NSCLC patients to determine the prognosis for them. For this purpose, the relationship between the parameters evaluating the stage of cancer and the expression of the studied genes in cancer tissue and blood was examined. The correlation between tissue and blood expression as well as the change in expression within one year after surgical removal of the pathological tissue was also assessed. 

The results of the analysis showed the presence of correlations between the expression of *MMP2*, *MMP9* and, interestingly, *TIMP1* in tumor tissue or blood. However, no correlation was found in the expression of the investigated genes between tissue and blood. This indicates a similar mechanism regulating the expression of these genes within the tissue, but not throughout the body (no correlation of expression was found between lung tissue and blood samples). Metalloproteinase and tissue metalloproteinase inhibitor genes are regulated by various types of cytokines, chemokines or growth factors [24]. Mitogen-activated protein kinase (MAPK), nuclear factor-κB (NFκB) and Smad-dependent pathways are involved in regulating the activation or inhibition of metalloproteinase gene expression [25]. The regulation of the expression of *MMP* and *TIMP* genes depends on the sequence of the promoter region. The presence of sequences containing canonical retinoic acid or retinoid X receptor response elements has also been demonstrated on the promoter regions of the *MMP2*, *MMP9* and *TIMP1* genes [26]. That could confirm the presence of correlations between the expression of the studied genes, but in the local environment only. Additionally, during the therapy, no changes in the expression of the tested genes in the blood were found, which may suggest that the change only affects the local environment of the tumor, while other tissues retain the correct structure of ECM transformation. The expression of *MMP2* and *MMP9* genes is also regulated at the level of post-transcriptional modifications or transcript stability [20]. An example is the miR-142 molecule, which affects the level of expression of both gelatinase genes, which has been proven in the case of osteosarcoma [27]. In the case of numerous solid tumors, there was an increase in the expression of both *MMP2* and *MMP9* genes which was associated with an unfavorable prognosis for patients [28]. The change in the expression of *MMP2* and *MMP9* concerned only cancer cells and no other cells. This may be due to the activation of the PI3K/Akt/mTOR pathway which is observed in NSCLC tumor cells [29]. Activation of this pathway may be associated with the increased expression of *MMP2* and *MMP9* observed in this disease entity [30].

The presented studies showed no effect of the adjuvant treatment on the expression of the *MMP2*, *MMP9* and *TIMP1* genes in the blood. This suggests a state in which even the use of chemotherapy does not cause a statistically significant change in the expression of the tested genes in the blood. Numerous studies indicate the influence of chemotherapy on the expression of genes of metalloproteinases or tissue inhibitors of metalloproteinases; however, they mainly concern changes involving neoplastic tissue. An example is cisplatin, which is widely used in NSCLC as an adjuvant therapy. Studies on tumor tissue fragments taken from patients before the initiation of treatment with cisplatin and vinorelbine and two weeks after the end of the treatment cycle showed a statistically significant decrease in *MMP2* gene expression compared to the baseline [31]. Similar results were also obtained by Liang H et al. who studied the effect of the use of cisplatin on the level of expression of MMP-2 and MMP-9 proteins in a human lung cancer cell line. The use of cisplatin was associated with lower levels of both tested proteins [32]. Interestingly, other studies evaluating the effect of cisplatin on *TIMP1* expression showed that its use significantly increased the expression of this inhibitor. The referenced test results describe a change at the level of cancer tissue or cancer cells, but they do not refer to changes in the blood, which only indicates a local change in the expression of metalloproteinase genes after the use of adjuvant chemotherapy [33].

Analysis of biological samples and information collected from external databases showed that the samples of the NSCLC patients were characterized by a higher gene expression of *MMP2* compared to normal tissue. Furthermore, higher expression of *MMP2* in tumor tissue samples was associated with shorter survival in patients with adenocarcinoma. Similar results were obtained for the second of the gelatinases, where the tumor tissue was also characterized by the expression of *MMP9* above the normal level. Again, a correlation between the level of *MMP9* gene expression and the length of survival of patients with adenocarcinoma was confirmed. Studies by Liping Han et al. [34], which compared MMP-2 protein levels between lung cancer and normal tissue samples by immunohistochemistry, showed higher protein levels in tumor tissues. Both cancer cells and surrounding normal cells were responsible for the synthesis of protein. Moreover, a positive correlation with cancer advancement parameters as well as with shorter survival of patients was also confirmed. Other zymographic studies assessing MMP-2 activity in lung cancer tissue samples compared to normal tissue showed higher lytic activity of cancer. Greater MMP-2 activity was associated with shorter patient survival [34]. Additionally, studies conducted by Ali-Labib et al. confirmed that patients with lung cancer have higher concentrations and activities of MMP-2 in serum and sputum compared to the control group [35]. Thus, it appears that MMP-2 plays a significant role in the development of lung cancer and might be used as a predictive marker of patient survival. As regards *MMP9*, El-Badrawy et al. confirmed the overexpression of the gene in NSCLC patients relative to controls. These studies were supplemented by a zymographic analysis of proteolytic activity, which also confirmed an increase in MMP-9-derived gelatinolytic activity in tissue samples from NSCLC patients [36]. Other studies conducted by Zhang et al. also confirmed the overexpression of the *MMP9* gene in NSCLC cells compared to normal cells. Moreover, a correlation was also observed between the higher MMP-9 activity and the increase in the potential for metastasis of cancer cells [37]. Furthermore, studies conducted by Wang et al. have also shown that the increase in MMP9 activity correlates with the growth of tumor metastatic potential [38]. Overexpression of *MMP2* and *MMP9* was associated with shorter overall survival in NSCLC patients. This may be due to the favorable role of both gelatinases in the progression of cancer. *MMP2* and *MMP9* participate both in the formation of new blood vessels and lymphangiogenesis, enabling adequate blood supply to the tumor, as well as the penetration of cancer cells into blood and lymphatic vessels. Moreover, both metalloproteinases are involved in the penetration of cells through the basement membrane, allowing the transfer of cancer cells to different sites of the body, giving rise to the process of metastasis [7]. Overexpression of *MMP2* and *MMP9* is associated with metastatic cell activity in many types of cancer [39]. Moreover, MMP-2 and MMP-9 impact the apoptosis process [7]. Most types of cancer cells overexpress TIMP-1 in relation to normal cells, which would seem to inhibit the neoplastic process. However, it promotes cell growth and proliferation and inhibits apoptosis.

An interesting study was an analysis of *TIMP1* expression carried out on available databases, i.e., UALCAN and Kaplan–Meier Plotter, which showed that tumor tissue fragments do not differ in expression from normal tissue. However, it also confirmed that *TIMP1* gene expression varied within tumor tissues, and higher expression was associated with shorter patient survival. This leads to the conclusion that *TIMP1* overexpression is beneficial for cancer development. Similar conclusions were drawn by Xiao et al., who observed a lower proliferative potential and increased apoptosis in non-small-cell lung cancer cells after silencing the *TIMP1* gene [40]. Results obtained by Simi et al. also confirmed the relationship between higher *TIMP1* expression and shorter survival in patients with NSCLC [41]. A high TIMP-1 protein expression level has been shown to be positively associated with poor prognosis or tumor progression. On the other hand, overexpression of each TIMP in tumor cell lines inhibits their migration, invasion, metastasis and subsequent growth. There are significant discrepancies between clinical and preclinical data, and the contribution of TIMPs in neoplastic disease remains unclear. The role of TIMP-1 is controversial, and it shows both pro- and anticancer effects [20].

The findings confirm that cancerous tissue in patients with NSCLC is characterized by a higher expression of *MMP2*, *MMP9* and, interestingly, *TIMP1*. Overexpression of *MMP2* and *MMP9* may be associated with increased angiogenesis, metastasis or lack of susceptibility to proapoptotic signals, while increased expression of *TIMP1* and other non-metalloproteinase inhibitors may promote excessive deposition of collagen fibers in lung tissue, thus leading to impaired pulmonary function and fibrosis [42].

## 5. Conclusions

In the presented study, the expression of *MMP2*, *MMP9* and *TIMP1* genes was assessed in the blood of NSCLC patients at three time points, before the tumor removal surgery, 100 days after surgery and one year after surgery, and in the tumor tissue. The study did not confirm the relationship between the level of expression of the studied genes and the length of survival of patients, which could be due to the insufficient number of patients included in the study. This resulted from the need to obtain a homogeneous group of patients who agreed to take blood samples at subsequent time points and appeared for the examination. However, evaluation of the results obtained from external databases confirmed the presence of a relationship between the level of expression of the *MMP2*, *MMP9* and *TIMP1* genes and the length of survival of patients. Work on the assessment of the role of the studied genes in NSCLC should also be continued at the level of assessing the concentration of individual proteins or assessing their activity.

The imbalance in the remodeling of the extracellular matrix accompanying the neoplastic process is manifested in changes in the activity of proteolytic enzymes degrading the ECM, e.g., metalloproteinases. To fully understand the process of carcinogenesis, it is necessary to study the changes that also occur in non-cellular structures. Restoring the balance between the production and degradation of the ECM may be an important aspect of therapy against cancer cells. However, inhibition of proteolytic properties alone does not seem to be a perfect solution, which was confirmed by clinical trials on metalloproteinase inhibitors. Perhaps a more targeted approach, taking into account the type of cancer, the stage of cancer or other clinical and demographical aspects, may be crucial.

## Figures and Tables

**Figure 1 biomedicines-11-01777-f001:**
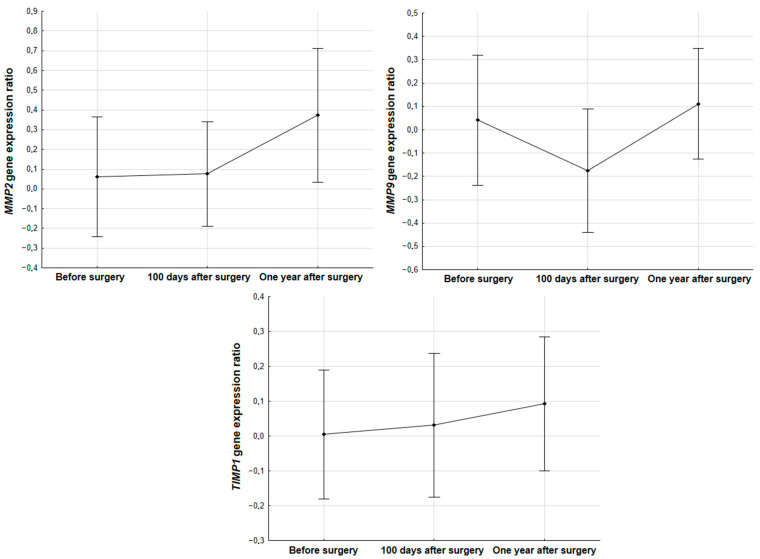
The relative expression level of *MMP2* (*p* = 0.1549), *MMP9* (*p* = 0.4667) and *TIMP1* (*p =* 0.1458) genes in the blood of patients with NSCLC before surgery, 100 days and one year after surgery.

**Figure 2 biomedicines-11-01777-f002:**
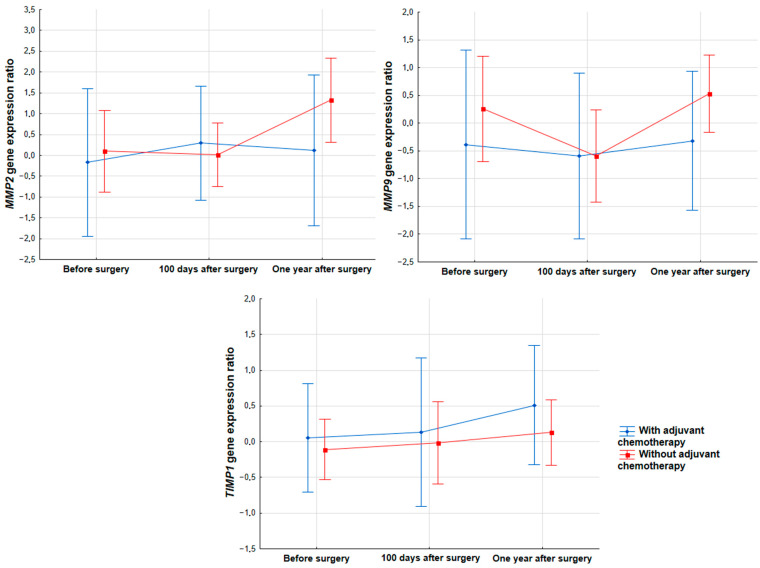
Expression of *MMP2* (*p* = 0.5377), *MMP9* (*p* = 0.7479) and *TIMP1* (*p* = 0.9051) genes in the blood of patients with NSCLC during therapy depending on adjuvant chemotherapeutic treatment.

**Figure 3 biomedicines-11-01777-f003:**
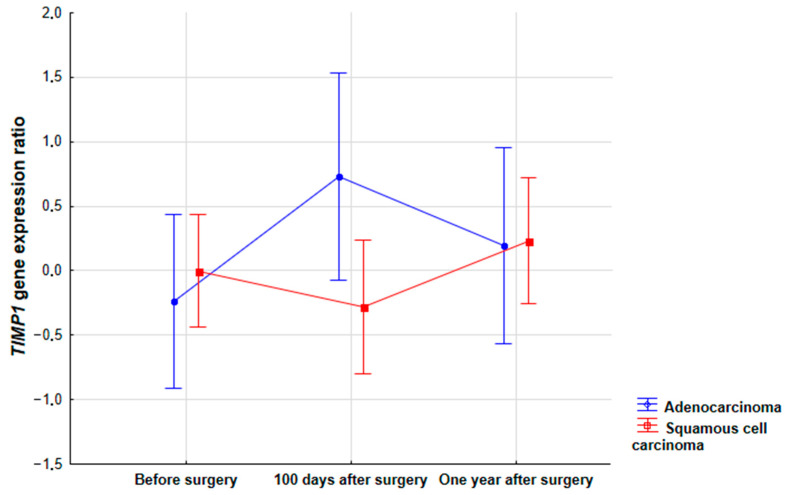
Comparison of *TIMP1* gene expression (*p* = 0.035) in the blood of patients with NSCLC during therapy based on the tumor histological type.

**Figure 4 biomedicines-11-01777-f004:**
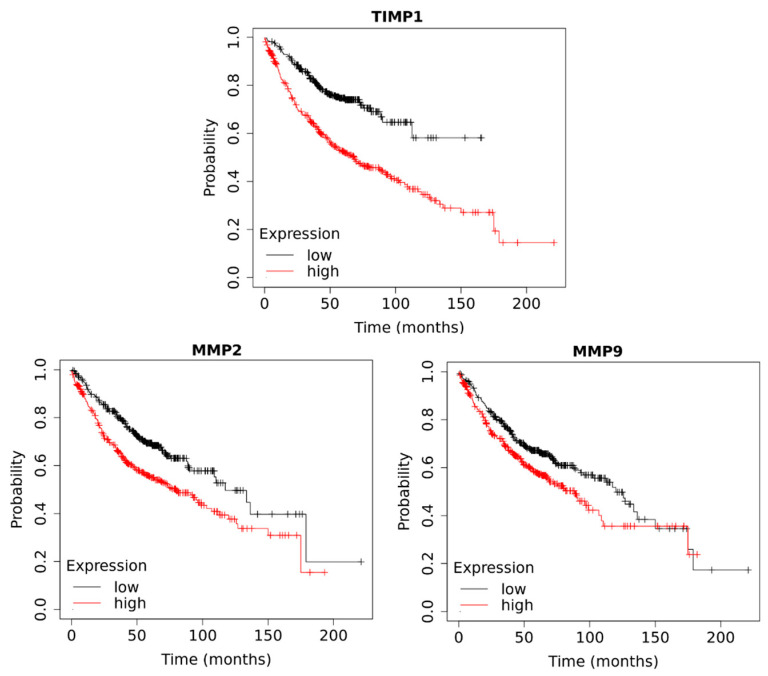
Kaplan–Meier plot comparing survival of patients with NSCLC (adenocarcinoma) depending on the expression of the *MMP2* (*p* < 0.001), *MMP9* (*p* = 0.009) and *TIMP1* (*p* < 0.001) genes in tumor tissue.

**Table 1 biomedicines-11-01777-t001:** Characteristics of the study group.

Clinical Parameter	Number of Patients		
Smoking tobacco products	Yes—32	No—13	
Gender	Men—35	Women—10	
Age	below median—23	above median—22	
Histological subtype	Adenocarcinoma—17	Squamous cell carcinoma—28	
Primary tumor size (T in TNM classification)	T1—13	T2—29	T3—3
Lymph node involvement (N in TNM classification)	N0—28	N1—8	N2—9
Presence of distant metastasis (M in TNM classification)	no distant metastasis—45		
Grading	G1—2	G2—32	G3—13

**Table 2 biomedicines-11-01777-t002:** Sequences of primer pairs used for PCR and qPCR reactions.

Primer Name	Sequence
*GAPDH* forward	5′-TGGTATCGTGGAAGGACTCATGAC-3′
*GAPDH* reverse	5′-ATGCCAGTGAGCTTCCCGTTCAGC-3′
*TIMP1* forward	5′-ATGCCAGTGAGCTTCCCGTTCAGC-3′
*TIMP1* reverse	5′-CACCTTATACCAGCATTATG-3′
*MMP2* forward	5′-TTTCCAGCAATGAGAAACTC-3′
*MMP2* reverse	5′-GTATCTCCAGAATTTGTCTCC-3′
*MMP9* forward	5′-CTTAGATCATTCCTCAGTGC-3′
*MMP9* reverse	5′-CGAGGACCATAGAGGTG-3′

**Table 3 biomedicines-11-01777-t003:** *P* Values and correlation coefficients of *TIMP1*, *MMP2* and *MMP9* expression in neoplastic tissue and blood of the NSCLC patients and the control group.

Group	Genes	*p*	r
NSCLC patients	***MMP2 blood* x *MMP2 tissue***	*0.530*	*-*
	***MMP9 blood* x *MMP9 tissue***	*0.487*	*-*
	***TIMP1 blood* x *TIMP1 tissue***	*0.992*	*-*
	***MMP2 blood* x *MMP9 blood***	** *<0.0001* **	** *0.6602* **
	***MMP2 blood* x *TIMP1 blood***	** *0.0150* **	** *0.3906* **
	***MMP9 blood* x *TIMP1 blood***	** *0.385* **	** *-* **
	***MMP2 tissue* x *MMP9 tissue***	** *<0.0001* **	** *0.7643* **
	***MMP2 tissue* x *TIMP tissue***	** *<0.0001* **	** *0.8385* **
	***MMP9 tissue* x *TIMP tissue***	** *0.0001* **	** *0.508* **
Control group	***MMP2 blood* x *MMP9 blood***	** *<0.0001* **	** *0.6897* **
	***MMP2 blood* x *TIMP1 blood***	** *0.0120* **	** *0.4039* **
	***MMP9 blood* x *TIMP1 blood***	** *0.0130* **	** *0.398* **

## Data Availability

The data presented in this study are available on request from the corresponding author.

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
