# Peer review of "Correlation of TIMP1-MMP2/MMP9 Gene Expression Axis Changes with Treatment Efficacy and Survival of NSCLC Patients"

_biomedicines, 2023, doi:10.3390/biomedicines11071777_

Round 1

Reviewer 1 Report

This manuscript investigates the effect of TIMP1-MMP2/MMP9 axis changes on treatment efficacy and survival of NSCLC patients. This is an interesting research paper and suit the aims and scope of the journal. However, there are some minor and major issues that need to be addressed prior to the acceptance.

Minor:

1) There numerous of sentences that are very misleading and difficult to understand.

2) The x and y axis need to be redrawn.

3) Please include the references in the materials and method section 

Major: 

1) The authors reported that higher expression of MMP2 and MMP9 was found in cancerous tissue but not in blood. This finding needs to be explained in the discussion. The authors could provide hypothesis on the differential expression of MMP2 and MMP9 in cancerous tissue and how these could be used in future for lung cancer treatment/detection.

2) The authors should include references that indicated the presence of MMP2, MMP9 and T1MP1 in lung cancer especially for NSCLC.

3) The authors should strengthen the discussion on the correlation between the impact of MMP2, MMP9 and T1MP1 in lowering the survival rate of NSCLC patients (Line 403-420).

4) The authors should discuss the impact of adjuvant presence during chemotherapy on the expression of MMP2, MMP9 and T1MP1. If possible, correlate the data to survival rate of NSCLC patients.

Need improvement.

Reviewer 2 Report

This study was aimed to evaluate the potential of MMP2, MMP9 and TIMP1 as prognostic markers for NSCLC. Gene expression levels of MMP2, MMP9 and TIMP1 in plasma and tumor tissues were determined in 45 patients with NSCLC. No correlation was found between gene expression in plasma and that in tumor tissue samples.  Expression of MMP2, MMP9 and TIMP1 genes in plasma and tumor tissues was not correlated with either tumor size or histological grade of malignancy. The analysis of data from the UALCAN database showed that the expression of the MMP2 and MMP9 (p<0.0001) genes in NSCLC tissues was significantly higher than that in the noncancerous tissues, and TIMP1 gene expression was inversely correlated with the survival of patients with lung cancer.

The lack of correlation of gene expression with either tumor size or histological grade of malignancy was likely due to the limited sample size, which undermined the validity of the study. In this regard, the overall merit of this study is low. 

Moderate editing of English language is needed. 

Round 2

Reviewer 2 Report

The authors have addressed all of my questions. 

Author Response

Thank you very much for your time and substantive comments on the manuscript.